# Household characteristics associated with surface contamination of SARS-CoV-2 and frequency of RT-PCR and viral culture positivity–California and Colorado, 2021

Talya Shragai[1,2☯], Caroline Pratt[1,2☯]*, Joaudimir Castro Georgi[1], Marisa A. P. Donnelly[1,2], Noah G. Schwartz[1,2], Raymond Soto[1,2], Meagan Chuey[1,2], Victoria T. Chu[1,2], Perrine Marcenac[1,2], Geun Woo Park[1], Ausaf Ahmad[1], Bernadette Albanese[3], Sarah Elizabeth Totten[4], Brett Austin[5], Paige Bunkley[1], Blake Cherney[1], Elizabeth A. Dietrich[1], Erica Figueroa[1], Jennifer M. Folster[1], Claire Godino[1], Owen Herzegh[1], Kristine Lindell[1], Boris Relja[1], Sarah W. Sheldon[1], Suxiang Tong[1], Jan Vinjé[1], Natalie J. Thornburg[1], Almea M. Matanock[1], Laura J. Hughes[1], Ginger Stringer[4], Meghan Hudziec[4], Mark E. Beatty[5], Jacquelin E. Tate[1], Hannah L. Kirking[1], Christopher H. Hsu[1], the COVID-19 Household Transmission Team[¶]

1 Centers for Disease Control and Prevention, Atlanta, Georgia, United States of America, 2 Epidemic Intelligence Service, Centers for Disease Control and Prevention, Atlanta, Georgia, United States of America, 3 Tri-County Health Department, Adams, Arapahoe, and Douglas Counties, Colorado, United States of America, 4 Colorado Department of Public Health and Environment, Glendale, Colorado, United States of America, 5 Health and Human Services, San Diego County, California, United States of America

☯ These authors contributed equally to this work.
¶ Membership of the COVID-19 Household Transmission Team is provided in the Acknowledgments.
* yqg5@cdc.gov

**Data Availability Statement:** All relevant data are within the paper and its Supporting Information files.

## Abstract

While risk of fomite transmission of SARS-CoV-2 is considered low, there is limited environmental data within households. This January—April 2021 investigation describes frequency and types of surfaces positive for SARS-CoV-2 by real-time reverse transcription polymerase chain reaction (RT-PCR) among residences with ≥1 SARS-CoV-2 infection, and associations of household characteristics with surface RT-PCR and viable virus positivity. Of 1232 samples from 124 households, 27.8% (n = 342) were RT-PCR positive with nightstands (44.1%) and pillows (40.9%) most frequently positive. SARS-CoV-2 lineage, documented household transmission, greater number of infected persons, shorter interval between illness onset and sampling, total household symptoms, proportion of infected persons ≤12 years old, and persons exhibiting upper respiratory symptoms or diarrhea were associated with more positive surfaces. Viable virus was isolated from 0.2% (n = 3 samples from one household) of all samples. This investigation suggests that while SARS-CoV-2 on surfaces is common, fomite transmission risk in households is low.

**Funding:** The authors received no specific funding for this work.

**Competing interests:** The authors have declared that no competing interests exist.

## Introduction

Since the beginning of the Coronavirus disease (COVID-19) pandemic in late December 2019, transmission of severe acute respiratory syndrome coronavirus 2 (SARS-CoV-2), the virus that causes COVID-19, has been a focus of public health prevention efforts worldwide [1]. While SARS-CoV-2 is spread mainly through respiratory droplets and aerosols, and transmission risk of SARS-CoV-2 from contaminated surfaces is considered low, it is difficult to differentiate between multiple simultaneous transmission pathways [2]. Previous studies detecting SARS-CoV-2 virus on surfaces have been conducted in healthcare [3–7] and community [4, 8] settings, and most published studies examining the viability of SARS-CoV-2 on surfaces have been conducted in controlled laboratory environments [9]. One study evaluated surface contamination of SARS-CoV-2 within households and showed that viable virus can be isolated from surfaces [10], but the small sample size of 150 samples limited that study's ability to assess household variables associated with contamination and frequency of surface contamination.

In this investigation, the frequency and types of surfaces contaminated with SARS-CoV-2 in households with one or more persons with SARS-CoV-2 infection are described. Surface swabs underwent RT-PCR testing to detect SARS-CoV-2 and isolation of viable virus by viral culture [11, 12]. Factors associated with RT-PCR detection of virus on household surfaces were explored including SARS-CoV-2 lineage, symptoms of infected household members at time of sample collection, time of surface swabbing since illness onset, household characteristics, and reported household prevention strategies.

## Materials and methods

### Household recruitment and enrollment

This investigation was embedded within a larger household transmission investigation. Recruitment and enrollment have been described in detail [13]. Briefly, the US Centers for Disease Control and Prevention (CDC) partnered with state and local public health departments in San Diego County, CA and Tri-County (Adams, Arapahoe, and Douglas Counties), CO to recruit households for participation. Households were enrolled by convenience sampling among households of all positive cases reported to the local health department per respective state from January 27 –April 1, 2021 in San Diego County and March 22 –April 16, 2021 in Tri-County. Primary cases were defined as the household member with earliest illness onset ≤10 days before enrollment. Secondary cases were household contacts with a positive RT-PCR for SARS-CoV-2 or seroconverted during the investigation without vaccination. Primary and secondary case classifications were assigned retroactively. Households in congregate settings, with multiple primary cases, lost to follow-up or withdrew, or having no enrolled primary cases or no environmental sampling at enrollment (Day 0) were excluded (Fig 1). Primary cases hospitalized or with illness onset >10 days from Day 0, were excluded. Each household was enrolled for 15 days (Days 0 to 14).

### Specimen collection and survey data

CDC investigators visited households at enrollment (Day 0) where each enrolled household member had nasopharyngeal (NP) swabs and blood specimens for serosurvey collected and completed questionnaires capturing demographics, symptoms and exposures to COVID-19. A household-level questionnaire was completed on Day 0 collecting household-level characteristics and COVID-19 mitigation behaviors. All household characteristics were evaluated at time of environmental sampling. All household members were asked to keep daily symptom diaries, during the enrollment. If any new household members developed COVID-19 symptoms or

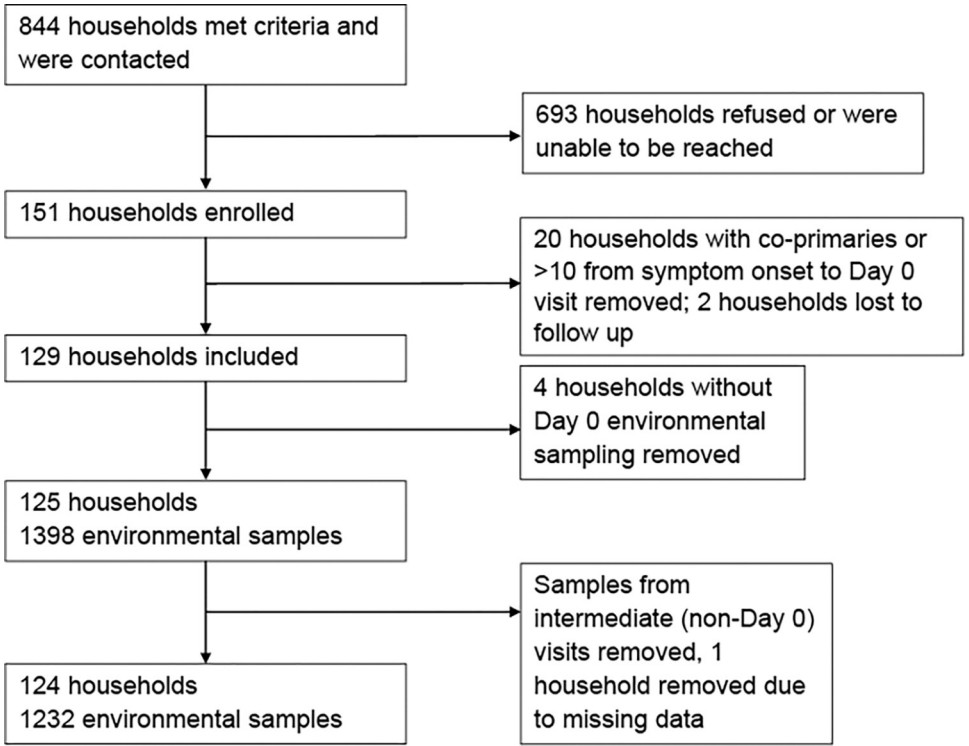

**Fig 1. Recruitment, enrollment, and exclusion of households and environmental samples.** Of 844 contacted households meeting study criteria, 151 households were enrolled. Households with co-primaries or >10 days from symptom onset to Day 0 visit were removed. Two households were lost to follow up. Only households with environmental samples and only environmental samples from Day 0 were included in the final analysis.

had a positive home antigen test during the enrollment period, investigators visited within 24 hours where NP specimens were collected from all enrolled household members. All households received a Day 14 closeout visit where NP and blood specimens were collected from enrolled household members. A more detailed description of all data and specimens collected as part of the larger study in which this study is embedded are described in Donnelly et al. [13].

Environmental swabs were collected using a previously validated and standardized procedure [10, 14]. Briefly, enrolled households voluntarily participated in environmental sampling at Day 0 [10]. In addition to swabbing at the initial visit, environmental sampling was offered to a subset of households at both interim and Day 14 closeout visits when sufficient swabs were available. Sani-MacroSwabs® (Sanigen, Anyang, Gyeonggi-do, South Korea) in vials filled with 7.5 mL of phosphate buffered solution were used to collect 10 samples from each consenting household. The following surfaces were sampled in each household on a 10 x10 cm square area: pillow or nightstand of index case, handle of toilet in the bathroom used by index case, faucet handle of bathroom sink used by index case, handle of refrigerator door, two commonly used light switches, the pillow or nightstand of two additional household contacts selected at the discretion of the investigator, a high-traffic area (e.g., kitchen counter), and a high-touch object (e.g., phone, remote, or doorknob). If a specific sample could not be collected (e.g., the household only had one additional household contact), a different sample was collected, and details of the collection surface were noted. Swabs were kept in coolers at 4˚C during the household visit for ≤4 hours and then stored in the local public health laboratory freezer at −80˚C. Specimens were shipped on dry ice to CDC for processing and analysis.

## Laboratory testing

**Nasopharyngeal swab processing and testing.** RT-PCR testing of NP swabs for SARS-CoV-2 were performed by the Colorado Department of Public Health and Environment (CDPHE) using the TaqPath™ COVID-19 Combo Kit (ThermoFisher Scientific) and the San Diego County Public Health Laboratories (SD PHL) using the New Coronavirus Nucleic Acid Detection Kit (PerkinElmer). NP swabs with cycle threshold (CT) values <35 were considered positive for RT-PCR.

NP specimens with CT values <35 underwent whole genome sequencing at CDC on the MinION platform (Oxford NanoPore Technologies) and the Illumina MiSeq platform (Illumina Inc.) or CDPHE on the GridION (Oxford NanoPore Technologies) or Illumina (Illumina Inc.) platform. SARS-CoV-2 sequences were assigned Phylogenetic Assignment of Named Global Outbreak Lineages (PANGOLIN) and a WHO label.

**Environmental swab processing and testing.** Nucleic acid extraction from environmental swabs was described previously [10] with the following modifications: approximately 4 mL of swab supernatant was transferred to an Amicon Ultra-4 filter device (30,000 MWCO, Millipore Sigma, Burlington, MA) and concentrated to 250 μL by centrifugation at 4,000× g for 10–20 minutes. The concentrated swab eluents were further processed for RNA extraction as described below.

Nucleic acid extraction of environmental surface samples was performed with the Qia-Cube-HT extraction system (Qiagen, Germantown, MD), using the QiaAmp 96 Virus kit (Qiagen, Germantown, MD) and QiaCube Plasticware according to the manufacturer's recommendations as previously described in McKay, et al. (2021) [15]. Extracted nucleic acid was tested for SARS-CoV-2 using the CDC influenza SARS CoV-2 (flu SC2) multiplex assay on the applied biosystems 7500 Fast DX real time PCR instrument (Thermo Fischer Scientific, Waltham, MA). The threshold for a positive result was a CT value ≤40.

Environmental RT-PCR swabs with CT values ≤28 were further tested by cell culture on Vero E6 / TMPRSS2 cells [11, 12]. Briefly, thawed concentrated samples were raised to 1.5 ml total volume with serum-free DMEM and filtered using a 0.45 μm syringe filter unit. Two hundred μl filtered samples were diluted 1:2 serially and used to inoculate adherent Vero E6 / TMPRSS2 cells. The remaining sample was used to inoculate adherent Vero E6 / TMPRSS2 cells in a T25 flask. Cultures were observed daily for cytopathic effects until seven days post-inoculation. Any cultures exhibiting cytopathic effect underwent confirmatory SARS-CoV-2 RT-PCR, using the same method as pre-culture RT-PCR. The sample was deemed culture positive if the CT value was at least 2.0 lower than the original sample. Based on previous literature [16–18] showing that virus has a low likelihood to be cultured when RT-PCR CT value is >28, specimens with CT >28 were not cultured and were combined for analysis with those testing negative by viral culture.

## Data analysis and statistical methods

Epidemiologic questionnaires were entered and stored in a REDCap database, version 10.0.8, (Nashville, TN) and laboratory results were entered and stored in a secure SharePoint drive (Redmond, WA), both hosted at CDC. All data analyses were performed in RStudio, version 4.0.4 (R Core Team, Boston, MA) or SAS statistical software (version 9.4, SAS Institute).

Day 0 surface swabs were classified into 17 categories based on household location from which they were collected. Frequencies and proportions of RT-PCR results were calculated for each category. CT value and RNA copy number median and interquartile range were calculated for positive samples by category. RNA copy numbers were log transformed.

Household variables hypothesized to be associated with household environmental surface contamination were selected for analysis based on literature review. Illness onset date was

defined as symptom onset date, or, if asymptomatic, collection date of first positive SARS-CoV-2 RT-PCR test. Household transmission was considered to have occurred if any household members tested positive or seroconverted after the primary case's first positive RT-PCR test and before the end of the Day 14. All variables referring to a frequency of use were measured on a four—point Likert scale from "never" to "always". Total household count of COVID-19 symptoms was assessed via a questionnaire asking each enrolled person if they were currently experiencing any of a list of 14 common COVID-19 symptoms. Participants responded either "yes" or "no" for each. The total count of "yes" responses was tallied for all household members for the day of environmental sampling. Proportion of ill household members that were children ≤12 years old was assessed by dividing number of positive household members that were ≤12 years old by total positive household members. Categorization of household lineages were based on lineages identified by the primary case, or if lineage of the primary case was unavailable, lineage of available secondary case. Household variants were grouped based on prevalence of circulating variants in communities at the time: B.1.1.7 (Alpha), B.1.427/B.1.429 (Epsilon) or other.

Univariate associations between categorical household characteristics (if one or more household members was experiencing a cough, runny nose and/or congestion, or diarrhea on day of sampling, if household transmission occurred, the variant of the primary case, the study site, household frequency of use of fans, heating, air conditioning, and windows, frequency of cleaning household surfaces, and if either ill or all household members wore masks) were assessed using ANOVA comparing the mean number of RT-PCR positive surface samples. Univariate associations of household characteristics utilizing continuous variables (total household count of COVID-19 symptoms on day of sampling, proportion of ill household members that were children ≤12 years old, number of household members testing positive on day of sampling, and days since most recent household member first tested positive) were analyzed by linear regression. Alpha of <0.05 was considered statistically significant.

A sensitivity analysis was conducted among interim and Day 14 environmental swabs, comparing results of univariate analyses with secondary sampling dates included. Because results did not change in the sensitivity analysis, results are presented with secondary sampling dates excluded. Eight households only had nine environmental surface samples taken, and similarly, a sensitivity analysis comparing results excluding households with nine environmental surfaces sampled was conducted. Because variables with a statistically significant result did not change in the sensitivity analysis, analyses are presented with those households included.

The percentage of swabs that were positive by viral culture was calculated for all RT-PCR positive samples and among all surface samples taken regardless of RT-PCR result.

### Ethical considerations

Adult participants provided written consent for enrollment; participants who were minors over the age of 7 years provided assent with parental consent, and participants under the age of 7 had parental consent provided. This activity was reviewed by CDC and was conducted consistent with applicable federal law and CDC policy (CDC ethics policy: 45 C.F.R part 46, 21 C. F.R. part 56; 42 U.S.C. §241(d); 5 U.S.C. §552a; 44 U.S.C. §3501 et seq).

## Results

### Households and surface samples included in analysis

A total of 125 households received environmental swabbing; 124 were analyzed and one was dropped due to missingness (Fig 1, Table 1). A total of 1232 samples from 124 households were analyzed (mean samples per household 9.9, range 9–10), of which 342 tested positive

**Table 1. Characteristics of households included in environmental sampling analysis.**

| Household Characteristic | Count and percentage of households N = 124, n (%) |
|---|---|
| **Site** | |
| CO | 71 (57.3) |
| CA | 53 (42.7) |
| **Variant** | |
| Alpha | 68 (54.8) |
| Epsilon | 16 (12.9) |
| Other | 23 (18.6) |
| Unable to sequence | 17 (13.7) |
| **Number of enrolled individuals per household** | |
| 2 | 34 (27.4) |
| 3 | 26 (21.0) |
| 4 | 31 (25.0) |
| 5 | 24 (19.4) |
| 6 | 6 (4.8) |
| 7 | 1 (0.8) |
| 8 | 2 (1.6) |
| **Did household transmission occur** | |
| Yes | 70 (56.5) |
| No | 54 (43.5) |
| **Percentage of household members testing positive during the study period** | |
| 0–25% | 17 (13.7) |
| 25.1–50% | 49 (39.5) |
| 50.1–75% | 19 (15.3) |
| 75.1–100% | 39 (31.5) |

(27.8%) by RT-PCR. Median CT value of positive samples was 34.0, with a range of 18.4–39.2 (Table 2). Of 124 households analyzed, ≥1 RT-PCR SARS-CoV-2 positive sample was recovered in 101 (85.3%) households, while 23 (14.7%) did not have any RT-PCR positive samples. A median of three positive surface samples was recovered per household, with an IQR of 1–4. Household transmission occurred in 56.5% (n = 70) of households.

Among variants identified here, 54.8% (n = 68) of households and 54.9% (n = 676) of environmental samples were collected from households where Alpha variant was detected in NP swabs, 12.9% (n = 16) of households and 13.0% (n = 160) environmental samples had the Epsilon variant, and 18.5% (n = 23) of households and 18.6% (n = 229) of environmental samples had other variants (Table 1). The variant was unable to be identified in 13.7% (n = 17) of households and 13.6% (n = 167) of environmental samples.

## Frequency of surface contamination by household surface type by RT-PCR

Light switches (n = 230, 18.6%), pillows (n = 176, 14.3%), and faucets (n = 148, 12%) made up the largest number of surface samples. Sixteen samples were categorized as "Other" (Table 2).

Nightstands were most frequently positive (n = 37, 44.1%), followed by pillows (n = 72, 40.9%), personal electronics (n = 24, 32.4%), counter tops (n = 13, 27.1%), and toilet handles (n = 34, 28.1%) (Table 2). Of positive surface samples, the lowest median CT values (median = 31.8, range = 23.6–36.8) and greatest log of RNA copy numbers (median = 2.5, range = 0.9–5.1) were from personal electronics. Samples collected from baby items, kitchen

**Table 2. SARS-CoV-2 RT-PCR results from environmental sampling of common household surfaces in households with one or more cases of COVID-19.**

| Surface | Total (n, %) | Positive RT-PCR (n, %) | Negative RT-PCR (n, %) | Inconclusive (n, %) | Median CT value (range)* | Median Log of RNA Copy Number (range) |
|---|---|---|---|---|---|---|
| Light switch | 229 (18.6) | 53 (23.1) | 176 (76.9) | 0 (0) | 34.1 (22.2–36.9) | 1.7 (1.1, 5.5) |
| Pillow | 176 (14.3) | 72 (40.9) | 103 (58.5) | 1 (0.6) | 33.7 (24.9–39.2) | 1.8 (0.5, 4.8) |
| Faucet | 148 (12.0) | 33 (22.3) | 115 (77.7) | 0 (0) | 34.6 (27.9–38.2) | 1.7 (0.8, 3.5) |
| Refrigerator | 124 (10.1) | 23 (18.6) | 100 (80.7) | 1 (0.8) | 34.3 (28.2–37.2) | 1.7 (1.0, 3.5) |
| Toilet handle | 121 (9.8) | 34 (28.1) | 87 (71.9) | 0 (0) | 34.3 (30.7–37.7) | 1.7 (0.9, 2.6) |
| Nightstand | 84 (6.8) | 37 (44.1) | 47 (56) | 0 (0) | 33.9 (20.8–37.9) | 1.8 (0.8, 5.5) |
| Doorknob/ handle | 80 (6.5) | 18 (22.5) | 62 (77.5) | 0 (0) | 34.0 (28.0–36.4) | 1.7 (1.2, 3.6) |
| Personal electronic | 74 (6.0) | 24 (32.4) | 50 (67.6) | 0 (0) | 31.8 (23.6–36.8) | 2.5 (0.9, 5.1) |
| Remote control | 68 (5.5) | 17 (25) | 51 (75) | 0 (0) | 34.1 (21.5–37.2) | 1.9 (1.1, 5.5) |
| Counter | 48 (3.9) | 13 (27.1) | 34 (70.8) | 1 (2.1) | 33.6 (18.4–38.3) | 2.0 (0.7, 6.4) |
| Furniture | 20 (1.6) | 7 (35) | 13 (65) | 0 (0) | 34.5 (27.6–37.0) | 1.7 (1.1, 3.6) |
| Other† | 16 (1.3) | 5 (31.3) | 10 (62.5) | 1 (6.3) | 33.5 (32.5–35.3) | 2.1 (1.3, 2.6) |
| Microwave | 13 (1.1) | 2 (15.4) | 11 (84.6) | 0 (0) | 33.1 (30.5–35.7) | 2.3 (1.5, 3.1) |
| Toy | 10 (0.8) | 3 (30) | 7 (70) | 0 (0) | 34.4 (33.9–34.8) | 1.6 (1.2, 1.8) |
| Kitchen item | 8 (0.6) | 0 (0) | 8 (100) | 0 (0) | NA | NA |
| Railing | 6 (0.5) | 0 (0) | 6 (100) | 0 (0) | NA | NA |
| Gate | 4 (0.3) | 1 (25) | 3 (75) | 0 (0) | 34.8 (34.8–34.8) | 1.7 (1.7, 1.7) |
| Baby item§ | 3 (0.2) | 0 (0) | 3 (100) | 0 (0) | NA | NA |
| **Total** | **1232** | **342 (27.7)** | **886 (71.9)** | **4 (0.003)** | **34.0 (18.4–39.2), 342** | **1.76 (0.5, 6.4)** |

*Positive samples were included in the calculation for median CT values.

†Items in "other" category included various items such as "trash can handle", "thermometer handle," and "shower door."

§Baby items included a crib, changing pod, and highchair.

appliances, and railings did not yield any positive samples and therefore did not have median CT or RNA copy number values.

## Association of household variables with surface contamination

The number of RT-PCR positive environmental surface samples was higher in households where infected persons reported more symptoms (p-value = 0.01) (Tables 3 and 4). The mean number of positive environmental surface samples was higher in households with ≥1 person with a cough on day of sampling compared to households with no members experiencing a cough (p-value = 0.005). Households with ≥1 person with a runny nose and/or congestion and households with ≥1 person with diarrhea had higher mean numbers of positive surface samples (p-value = 0.03; p-value = 0.04, respectively).

Household transmission was associated with recovering more positive surface samples (p-value<0.0001), as were households infected with Epsilon variant(p-value = 0.01). When the proportion of infected persons in the household represented by children ≤12 years-old increased, the average number of RT-PCR positive environmental surface samples increased (p-value = 0.01), but the proportion of infected household members that were between 12 and 18 years old or ≥18 years old was not statistically significantly correlated to the number of positive environmental surface samples (p-value = 0.2, p-value = 0.6, respectively). The greater the number of people testing RT-PCR positive for SARS-CoV-2 by NP swab on day of testing, the more positive surface samples were recovered on average (p-value<0.0001). As the number of

**Table 3.** Household characteristics, total environmental samples collected, and univariate analysis of association between categorical household characteristics and number of positive environmental samples.

| Household characteristic[*] | Samples collected N = 1232, n | Mean number of samples testing RT-PCR positive (IQR) | p-value[†] |
|---|---|---|---|
| **One or more household members is experiencing a cough** | | | **0.005** |
| *Yes* | 835 | 3.1 (1–4) | |
| *No* | 338 | 1.8 (0–3) | |
| *Missing* | 59 | NA | |
| **One or more household members is experiencing a runny nose and/or congestion** | | | **0.03** |
| *Yes* | 896 | 3.0 (1–4) | |
| *No* | 277 | 1.9 (0–3) | |
| *Missing* | 59 | NA | |
| **One or more household members is experiencing diarrhea** | | | **0.04** |
| *Yes* | 158 | 3.7 (2–5) | |
| *No* | 925 | 2.5 (1–4) | |
| *Missing* | 149 | NA | |
| Did household transmission occur[§] | | | **<0.0001** |
| *Yes* | 697 | 3.6 (2–5) | |
| *No* | 535 | 1.7 (0–3) | |
| **SARS-CoV-2 household variant** | | | **0.01** |
| *Epsilon* | 160 | 4.4 (2.8–5.3) | |
| *Alpha* | 676 | 3.0 (1–4) | |
| *Other* | 229 | 2.2 (1–3.5) | |
| *Unable to sequence* | 167 | NA | |
| **Study site** | | | 0.9 |
| *California* | 524 | 2.8 (0–4) | |
| *Colorado* | 708 | 2.7 (1–4) | |
| **Household use of fans** | | | 0.2 |
| *Never* | 595 | 3 (1–4) | |
| *Rarely* | 229 | 2 (1–4.5) | |
| *Sometimes* | 180 | 2.5 (1–3.8) | |
| *Always* | 228 | 2 (1–3.5) | |
| **Household use of heating** | | | 0.6 |
| *Never* | 416 | 2.7 (1–4) | |
| *Rarely* | 367 | 2.9 (1–5) | |
| *Sometimes* | 190 | 3.4 (1.5–4) | |
| *Always* | 249 | 2.2 (1–3) | |
| *Missing* | 10 | NA | |
| **Household use of air conditioning** | | | 0.8 |
| *Never* | 1042 | 2.7 (1–4) | |
| *Rarely* | 120 | 3.3 (1.5–3.5) | |
| *Sometimes* | 40 | 2.8 (0.75–4.5) | |
| *Always* | 20 | 2.5 (1.3–3.8) | |
| *Missing* | 10 | NA | |
| **Does household leave windows open** | | | 0.9 |
| *Never* | 239 | 2.8 (1–4) | |
| *Rarely* | 339 | 2.9 (1–4.8) | |
| *Sometimes* | 289 | 2.4 (1–4) | |
| *Always* | 365 | 3.0 (1–4) | |

(*Continued*)

**Table 3.** (Continued)

| Household characteristic* | Samples collected N = 1232, n | Mean number of samples testing RT-PCR positive (IQR) | p-value† |
|---|---|---|---|
| **Does household clean/sanitize surfaces** | | | 0.8 |
| *Never* | 100 | 3.0 (1–3.8) | |
| *Rarely* | 328 | 2.7 (1–4) | |
| *Sometimes* | 309 | 3.0 (1–6) | |
| *Always* | 475 | 2.7 (1–4) | |
| *Missing* | 20 | NA | |
| **Do ill members of the household wear masks** | | | 0.5 |
| *Never* | 457 | 2.8 (1–4) | |
| *Rarely* | 178 | 3.3 (1.3–5) | |
| *Sometimes* | 199 | 2.7 (1–4) | |
| *Always* | 398 | 2.5 (1–4) | |
| **Do all members of the household wear masks** | | | 0.3 |
| *Never* | 546 | 2.6 (1–4) | |
| *Rarely* | 248 | 2.6 (1–4) | |
| *Sometimes* | 179 | 2.7 (1–4) | |
| *Always* | 259 | 3.3 (1–4.8) | |
| **Household square footage** | | | 0.9 |
| *<1300* | 267 | 2.7 (1–4) | |
| *1300–1999* | 306 | 2.5 (1–4.5) | |
| *2000–2999* | 249 | 2.5 (1–3) | |
| *≥3000* | 350 | 2.2 (1–4) | |
| *Missing* | 60 | NA | |

*All household characteristics were evaluated at time of environmental sampling.

†P-values were calculated by ANOVA comparing the mean number of RT-PCR positive surface samples by household characteristics. Analyses exclude missing values. Statistically significant results (p-value<0.05) in bold.

§ Household transmission was considered to have occurred in a household if any household members tested positive at any time point after the primary case's first positive RT-PCR test.

days between the most recent positive person's first positive test and day of surface sampling increased, the number of positive samples decreased (p-value<0.0001); however, days between the primary case's first positive test result and the day of surface sampling was not associated with number of positive surface samples (p-value = 0.2).

There was no statistically significant difference in mean number of positive samples in households by study site (p-value = 0.9). No frequency of any household mitigation had a statistically significant association with the number of household environmental surface samples testing positive by RT-PCR; this included frequency of cleaning household surfaces, frequency of using air conditioning, fans, heating, and opening windows, and frequency of either ill people or all household members wearing masks (all p-values >0.05). Household square footage was not statistically associated with number of positive household surface samples (p-value = 0.9).

## Frequency of recovery of viable virus

Of the 36 samples with CT value ≤28 submitted for viral culture by the CDC lab, 29 (median = 27.2, range = 18.4–29.1) met the inclusion criteria for this analysis. Viable SARS-CoV-2 virus was detected in three samples (CT values of 18.4, 20.8, and 21.5, respectively),

**Table 4. Household characteristics, and univariate analysis of association between continuous household characteristics and number of positive environmental samples.**

| Household characteristic* | Coefficient (95% Confidence Interval) | p-value† |
|---|---|---|
| Total number of symptoms experienced by all household members | 0.08 (0.02–0.1) | **0.01** |
| Proportion of household members testing RT-PCR positive that are children ≤ 12 years old§ | 2.2 (0.6–3.9) | **0.01** |
| Proportion of household members testing RT-PCR positive that are children >12 & <18 years§ old§ | -0.9 (-2.2–0.5) | 0.2 |
| Proportion of household members testing RT-PCR positive that are adults ≥18 years old§ | -0.3 (-1.5–0.8) | 0.6 |
| Number of household members testing RT-PCR positive for SARS-CoV-2 on day of environmental sampling | 0.6 (0.3–0.9) | **<0.0001** |
| Days since primary case received their first positive test result | -0.2 (-0.5–0.1) | 0.2 |
| Days since most recent household member testing positive received their first positive test result | -0.3 (-0.5–0.1) | **<0.0001** |

*All household characteristics were evaluated at time of environmental sampling.

†P-values were calculated by linear regression correlating the number of RT-PCR positive samples by household characteristics. Analyses exclude missing values. Statistically significant results (p-value<0.05) in bold.

§Proportion of household members testing RT-PCR positive that were either children ≤ 12, children >12 to ≤ 18, or adults >18 was calculated by dividing the count of positive household members in each age group by the total number of positive household members.

representing 0.9% of all RT-PCR positive samples and 0.2% of all surface samples taken. These three samples were recovered from a single household in Colorado. All ten surface samples in that household were positive by RT-PCR, and the three surfaces positive by viral culture were the primary case's nightstand, the remote control, and the kitchen counter. The household had two members; the primary case was a 50-year-old female, and the secondary case was an 18-year-old male, with no other household members residing in this household. The primary case first tested positive five days before environmental surface sampling, and the secondary case first tested positive on the day of surface sampling. Both were symptomatic on day of sampling, each reporting eight of 15 symptoms surveyed, including nasal symptoms and cough. Neither reported diarrhea. The primary case reported a history of hypertension, hyperlipidemia, and unspecified immunosuppression. The household was a 3-bed 3-bathroom between 2000–3000 square feet. Household members reported sometimes using AC, using heating most of the time, and sometimes opening windows for ventilation and using fans. The household reported never cleaning high-touch household surfaces. The Epsilon variant was characterized from NP specimens of both household members.

## Discussion

Pillows and nightstands were most commonly positive for SARS-CoV-2 by RT-PCR testing, confirming the findings of Marcenac, et al. [10]. As indicated by the high RT-PCR positivity from nightstands and pillows in sleeping areas of symptomatic COVID-19 cases, breathing (and possibly coughing) unmasked for several hours each night near these items likely contributed to high positivity. This could also support the recommendation to isolate asymptomatic confirmed cases given that surfaces in close proximity to cases and where cases spend several hours each day (and breathe for several hours each day) appear to be more likely to be positive for SARS-CoV-2 [19].

It is important to note that RT-PCR testing simply detects genetic material, not transmissible virus, as indicated by the low positive viral cultures from environmental swabs. This is consistent with research on other species of human coronavirus in the environment, which are usually not stable on surfaces [20]. Previous studies have shown that in laboratory conditions, SARS-CoV-2 survives ≤3 hours in the air, 4 hours on copper, 24 hours on cardboard, 48 hours on steel, and 72 hours on plastic [21].

Three samples positive for viable virus were collected from a single household; it is known that some individuals spread SARS-CoV-2 at higher rates than others [22, 23], and similarly some individuals may cause surface contamination with viable SARS-CoV-2 at higher rates. Additionally, one household member first tested positive for SARS-CoV-2 on the day of sampling, likely contributing to increased RT-PCR positivity of environmental samples. This household was positive for the Epsilon variant (a variant of concern at the time of investigation); some variants may shed more than others or have higher rates of surface survival [24]. These conclusions, however, are speculative due to limited sample size, and more research is needed.

Most houses included here had at least one surface sample test positive by RT-PCR, suggesting that viral shedding onto surfaces is common. Homes with household transmission and households infected with Epsilon variant had higher levels of surface contamination; this may be because some variants and transmission are associated with higher viral load and higher viral shedding [25]. Also, households with a greater proportion of infected persons 12 years or younger had higher levels of surface contamination. Most studies show that children tend to have equal or lower viral loads compared to adults [26]. Our findings may be due to behavioral characteristics of children such as increased hand to face contact [27] or other confounding factors not addressed here. Interestingly, many mitigating behaviors predicted to reduce surface contamination or known to reduce person-to-person transmission (e.g., cleaning high touch surfaces, increasing air ventilation, and wearing masks) [28, 29] were not associated with lower numbers of RT-PCR positive environmental samples.

This investigation had limitations. First, at time of data collection, the SARS-CoV-2 Alpha variant was the primary variant of concern in the United States, and it is unknown if these results can be generalized to Delta, Omicron, or other variants. Second, only 3 surface samples tested positive by viral culture, and all were from a single household. This sample size was too small to conduct a full analysis of factors associated with detection of culturable virus on surfaces which will be an important area for future studies. It is also impossible to assess if there was some variation in collection protocol that led to these results such as variable swab sampling efficacy or variation in swabbing surface area between data collectors. Third, sample collection may not have been timed with peak viral shedding among infected household members. Fourth, household surface samples cannot be traced to individuals or to primary vs secondary cases, and therefore correlating individual characteristics with levels of surface contamination was not possible within this study design. Fifth, this study was conducted in only two locations. Location-specific factors, such as climate or types of HVAC favored, may contribute to availability of recoverable RNA or culturable virus. Finally, the analyses of household characteristics associated with higher RT-PCR positive surface samples are purely exploratory and do not suggest causation. Associations here may be confounded by other factors or be proxies for other underlying causal variables.

This investigation extends previous findings on surface contamination of SARS-CoV-2 to household settings. While previous SARS-CoV-2 fomite studies focused on healthcare [3–7] and community [4, 8] settings, this is the largest investigation the authors are aware of that focused on households. While viral shedding onto surfaces is common, the risk of SARS-CoV-2 fomite transmission is likely low, and management of SARS-CoV-2 infections in households

should prioritize reducing airborne person-to-person transmission with both non-medical interventions and up-to-date vaccination. Based on our results, however, fomite transmission in households may be possible.

## Supporting information

**S1 File. B117_HH_master PLOS variables.**
(CSV)

**S2 File. Environmental data CA CO.**
(XLSX)

**S3 File. Swab data summary.**
(CSV)

## Acknowledgments

The authors acknowledge the following individuals for their valuable contributions on the COVID-19 Households Transmission Team: Anna Drexler, Aaron Brault, Janet McAllister, Janae Stovall, Jamie Pawloski, Karen Boroughs. The authors also acknowledge all the households who participated willingly in this study and thanks them for their participation.

**Author Bio** (first author only, unless there are only 2 authors).

Caroline Pratt and Talya Shragai are both Epidemic Intelligence Service Officers at the Centers for Disease Control and Prevention. Within this role their research has focused on various aspects of the COVID-19 response.

## Author Contributions

**Conceptualization:** Talya Shragai, Caroline Pratt, Perrine Marcenac, Jacqueline E. Tate, Hannah L. Kirking, Christopher H. Hsu.

**Data curation:** Talya Shragai, Caroline Pratt, Joaudimir Castro Georgi, Geun Woo Park.

**Formal analysis:** Talya Shragai, Caroline Pratt, Marisa A. P. Donnelly.

**Investigation:** Talya Shragai, Caroline Pratt, Joaudimir Castro Georgi, Marisa A. P. Donnelly, Noah G. Schwartz, Raymond Soto, Meagan Chuey, Victoria T. Chu, Perrine Marcenac, Geun Woo Park, Ausaf Ahmad, Bernadette Albanese, Sarah Elizabeth Totten, Brett Austin, Paige Bunkley, Blake Cherney, Elizabeth A. Dietrich, Erica Figueroa, Jennifer M. Folster, Claire Godino, Owen Herzegh, Kristine Lindell, Boris Relja, Sarah W. Sheldon, Suxiang Tong, Jan Vinjé, Natalie J. Thornburg, Almea M. Matanock, Laura J. Hughes, Ginger Stringer, Meghan Hudziec, Mark E. Beatty, Jacqueline E. Tate, Hannah L. Kirking, Christopher H. Hsu.

**Methodology:** Talya Shragai, Caroline Pratt, Joaudimir Castro Georgi, Marisa A. P. Donnelly, Noah G. Schwartz, Meagan Chuey, Victoria T. Chu, Mark E. Beatty, Jacqueline E. Tate, Christopher H. Hsu.

**Project administration:** Mark E. Beatty, Christopher H. Hsu.

**Supervision:** Jacqueline E. Tate, Hannah L. Kirking, Christopher H. Hsu.

**Writing – original draft:** Talya Shragai, Caroline Pratt, Joaudimir Castro Georgi.

**Writing – review & editing:** Talya Shragai, Caroline Pratt, Joaudimir Castro Georgi, Marisa A. P. Donnelly, Noah G. Schwartz, Raymond Soto, Meagan Chuey, Victoria T. Chu, Perrine

Marcenac, Geun Woo Park, Ausaf Ahmad, Bernadette Albanese, Blake Cherney, Erica Figueroa, Jennifer M. Folster, Suxiang Tong, Jan Vinjé, Almea M. Matanock, Laura J. Hughes, Mark E. Beatty, Jacqueline E. Tate, Hannah L. Kirking, Christopher H. Hsu.

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
