## [Decision Letter · Decision Letter 0]

8 Aug 2022

PONE-D-22-10062Household characteristics associated with surface contamination of SARS-CoV-2 and frequency of RT-PCR and viral culture positivity– California and Colorado, 2021.PLOS ONE

Dear Dr. Pratt,

Thank you for submitting your manuscript to PLOS ONE. After careful consideration, we feel that it has merit but does not fully meet PLOS ONE’s publication criteria as it currently stands. Therefore, we invite you to submit a revised version of the manuscript that addresses the points raised during the review process.

Specifically, reviewer 1 has suggested minor edits and raised some questions. Those comments should be addressed before publication.

We look forward to receiving your revised manuscript.

Kind regards,

Ricardo Santos

Academic Editor

PLOS ONE

Journal Requirements:

Additional Editor Comments (if provided):

Reviewers' comments:

Reviewer's Responses to Questions

**Comments to the Author**

1. Is the manuscript technically sound, and do the data support the conclusions?

Reviewer #1: Yes

Reviewer #2: Yes

2. Has the statistical analysis been performed appropriately and rigorously? 

Reviewer #1: Yes

Reviewer #2: Yes

3. Have the authors made all data underlying the findings in their manuscript fully available?

Reviewer #1: Yes

Reviewer #2: Yes

4. Is the manuscript presented in an intelligible fashion and written in standard English?

Reviewer #1: Yes

Reviewer #2: Yes

5. Review Comments to the Author

Reviewer #1: The paper contributes important information to the knowledge base on SARS-CoV-2 and possible fomite transmission and surface contamination. It is novel in that it included household level data and had a large enough sample to successfully conclude findings in an uncontrolled environment. The results of this paper should be repeated in order to conclude the same for other SARS-CoV-2 variants. The authors did an excellent job in explaining the methodology and ethical clearance. Enough attention is given to detail wrt the statistical analyses. The paper clearly explains their findings with proof from other similar studies and recent papers/ work were cited. English language use is clear and does not require specific editing or language improvement. The authors, in detail, highlighted the uncertainties and the need for further studies to improve this work.

Minor edits are suggested in track changes and the editor can go ahead with publishing while authors accept or reject these. I would strongly recommend an acknowledgement to those households who willingly participated in the study.

The reviewer was interested in the importance of the cleaning regime / timing (when last the house was cleaned) combined with size of households and number of individuals with symptoms. Correlation of these together with timing of testing with onset of illness would be interesting in future. If the main caregiver is ill and the household is not cleaned as regularly as always, it could possibly add to surface contamination.

One of the main questions / queries are the impact of asymptomatic carriers to the level of surface contamination. The reviewer is not clear if this has been dealt with in terms of the antigen test, however false negative test results should then be considered.

Congratulations on work well done. It is therefore recommended that this paper is published.

Reviewer #2: In this manuscript the authors investigated the possible role of surfaces contamination in SARS CoV2 transmission in households.

The topic is very interesting because few data are available on the chain of transmission in household comparing to other settings as healthcare ones. The obtained data are well described with deep statistical analysis, moreover the discussion is clear and well written, no corrections in the text are needed.

In my opinion the manuscript can be published in this journal.

6. PLOS authors have the option to publish the peer review history of their article (what does this mean?). If published, this will include your full peer review and any attached files.

Reviewer #1: No

Reviewer #2: No

---

## [Author Response · Author response to Decision Letter 0]

11 Aug 2022

Household characteristics associated with surface contamination of SARS-CoV-2 and frequency of RT-PCR and viral culture positivity – California and Colorado, 2021. 

Reviewer One

1. Minor edits are suggested in track changes and the editor can go ahead with publishing while authors accept or reject these. I would strongly recommend an acknowledgement to those households who willingly participated in the study.

All suggested edits have been reviewed and are acceptable to the authors. An acknowledgement to the participating households has been added on line 570-571. 

2. The reviewer was interested in the importance of the cleaning regime / timing (when last the house was cleaned) combined with size of households and number of individuals with symptoms. Correlation of these together with timing of testing with onset of illness would be interesting in future. If the main caregiver is ill and the household is not cleaned as regularly as always, it could possibly add to surface contamination.

The authors appreciate this comment; we agree that analyzing the correlation between timing of last cleaning and positive surface samples would be interesting and useful. Unfortunately, our analysis is limited to the survey questions asked, and we did not include any questions about the timing of the last cleaning. However, the survey question “How frequently are you cleaning household surfaces” is in reference to current practices while enrolled in our study rather than general practices, which gives us some idea of how ongoing cleaning while household members are sick or lack there-of correlates to surface sample positivity. This is included as a footnote on Table 3 (“All household characteristics were evaluated at time of environmental sampling”), and we have additionally added this clarification to the methods section on line 155-156. 

3. One of the main questions / queries are the impact of asymptomatic carriers to the level of surface contamination. The reviewer is not clear if this has been dealt with in terms of the antigen test, however false negative test results should then be considered.

We did not assess in detail the transmission from asymptomatic individuals on fomite surfaces; however, tables 3 and 4 indicate that households that were symptomatic had a statistically higher number of positive surface swabs detected by RT-PCR. Also, viable virus was isolated in only 3 swabs in our study. Taken together, our results indicate that fomite contamination is more likely to occur if individuals are symptomatic in a household but the risk of transmitting viable virus to a susceptible contact through fomites was very low. 

4. To clarify – was [environmental sampling] repeated? Was swabs taken at interim and again on day 14 within the same household once either at interim or on day 14? This could possibly explain some of the low positive test results.

We have clarified in the text, and line 187-189 has been revised to read “In addition to swabbing at the initial visit, environmental sampling was offered to a subset of households at both interim and Day 14 closeout visits when sufficient swabs were available.”

5. In reference to table 1, the reviewer has asked if we could include Household cleaning of surfaces as a table variable. 

This table is meant to summarize key household characteristics, not household behaviors. The distribution of households cleaning surfaces on a Likert scale is shown in table 3

6. Was any of the individual household’s data further investigated and were there any interesting aspects (even though the sample size was too small to conclude anything worth reporting)? Specifically, the day of onset, the inclusion of children younger than 12 etc. Was this household different in terms of size, cleaning, etc. 

Lines 409-419 have been revised to include additional detail: 

“The household had two members; the primary case was a 50-year-old female, and the secondary case was an 18-year-old male, with no other household members residing in this household. The primary case first tested positive five days before environmental surface sampling, and the secondary case first tested positive on the day of surface sampling. Both were symptomatic on day of sampling, each reporting eight of 15 symptoms surveyed, including nasal symptoms and cough. Neither reported diarrhea. The primary case reported a history of hypertension, hyperlipidemia, and unspecified immunosuppression. The household was a 3-bed 3-bathroom between 2000-3000 square feet. Household members reported sometimes using AC, using heating most of the time, and sometimes opening windows for ventilation and using fans. The household reported never cleaning high-touch household surfaces. The Epsilon variant was characterized from NP specimens of both household members.”

Reviewer Two

This reviewer did not make any suggested revisions.

---

## [Editor Report · Decision Letter 1]

8 Sep 2022

Household characteristics associated with surface contamination of SARS-CoV-2 and frequency of RT-PCR and viral culture positivity– California and Colorado, 2021.

PONE-D-22-10062R1

Dear Dr. Pratt,

We’re pleased to inform you that your manuscript has been judged scientifically suitable for publication and will be formally accepted for publication once it meets all outstanding technical requirements.

Kind regards,

Ricardo Santos

Academic Editor

PLOS ONE
---

## [Editor Report · Acceptance letter]

29 Sep 2022

PONE-D-22-10062R1 

Household characteristics associated with surface contamination of SARS-CoV-2 and frequency of RT-PCR and viral culture positivity– California and Colorado, 2021. 

Dear Dr. Pratt:

I'm pleased to inform you that your manuscript has been deemed suitable for publication in PLOS ONE. Congratulations! Your manuscript is now with our production department. 

Kind regards, 

on behalf of

Dr. Ricardo Santos 

Academic Editor

PLOS ONE